# *Flos populi* (Male Inflorescence of *Populus tomentosa* Carrière) Aqueous Extract Suppresses *Salmonella* Pullorum Infection by Affecting T3SS-1

**DOI:** 10.3390/pathogens12060790

**Published:** 2023-05-31

**Authors:** Wenting Zhang, Guixing Liang, Zhenyu Cheng, Yunqing Guo, Boda Jiang, Tingjiang Liu, Weidong Liao, Qin Lu, Guoyuan Wen, Tengfei Zhang, Qingping Luo

**Affiliations:** 1Key Laboratory of Prevention and Control Agents for Animal Bacteriosis (Ministry of Agriculture and Rural Affairs), Hubei Provincial Key Laboratory of Animal Pathogenic Microbiology, Institute of Animal Husbandry and Veterinary, Hubei Academy of Agricultural Sciences, Wuhan 430064, China; 2Department of Microbiology and Immunology, Dalhousie University, Halifax, NS B3H 4R2, Canada; 3Hubei Hongshan Laboratory, Wuhan 430064, China

**Keywords:** *Flos populi*, *Salmonella* Pullorum, anti-infection, T3SS-1

## Abstract

Pullorum disease, caused by *Salmonella* Pullorum (*S*. Pullorum), is one of the most serious infectious diseases in the poultry industry. *Flos populi* is traditionally used in Eastern Asian countries to treat various intestinal diseases. However, the anti-infection mechanism of *Flos populi* is not very clear. In this study, we evaluated the anti-infective effects on *S.* Pullorum of *Flos populi* aqueous extract (FPAE) in chickens. FPAE significantly reduced *S.* Pullorum growth in vitro. At the cellular level, FPAE reduced *S.* Pullorum adhesion and invasion on DF-1 cells but did not affect its intracellular survival or replication in macrophages. Further investigation revealed that FPAE inhibited the transcription of T3SS-1 genes, which is the main virulence factor that mediates *S.* Pullorum adhesion and invasion in host cells. The results suggest that the anti-infective effect of FPAE likely occurs through the inhibition of *S.* Pullorum T3SS-1, thereby impairing its ability to adhere to and invade cells. Further, we evaluated its therapeutic effect on animal models (Jianghan domestic chickens) and found that FPAE reduced the bacterial loads in organs and decreased the mortality and weight loss of infected chickens. Our findings provide novel insights into the potential development of FPAE against *S.* Pullorum as an effective anti-virulence therapeutic substitute for antibiotics.

## 1. Introduction

*Salmonella*, a Gram-negative, non-spore-forming facultative anaerobic bacillus, belongs to the family *Enterobacteriaceae* [1]. Salmonellosis is a major public health concern that causes human infections, endangers food safety, and results in severe economic losses. According to the White-Kauffmann-Le Minor scheme, there are only two species of *Salmonella* and six sub-species within the species enterica, but more than 2600 serovars [2]. *Salmonella enterica* serovar Gallinarum biovar Pullorum (*S*. Pullorum) is a pullorum agent that can cause high mortality in young chicks. Because of its vertical and horizontal transmission, as well as the fact that artificial incubation favors its spread, *S*. Pullorum is a threat to the modern poultry industry, causing great economic losses [3].

Antibiotics are frequently used to control salmonellosis in poultry. However, antibiotic resistance in *S.* Pullorum has been observed in many areas worldwide. Developing vaccines to prevent salmonellosis is challenging owing to the large number of serovars, which require different vaccines for different or related serovars of *Salmonella*. Therefore, appropriate drugs (preferably antibiotic replacements, such as anti-virulence agents) must be developed, and targets in the *Salmonella* pathogenic machinery must be identified [4]. One current research trend is the investigation of plant-origin medicines because they are affordable and have minimal side effects [5].

*Flos populi* (YangShuHua) is an important traditional Chinese medicine prepared from *Populus tomentosa* Carrière or *P. canadensis* Moench (Salicaceae) [6]. Chinese white poplar bark extracts were prescribed to treat pediatric scalp favus, enteritis, diarrhea, and dysentery in the compendium of materia medica. In Eastern Asian nations, *Flos populi* is currently primarily used to treat a variety of inflammatory diseases and diarrhea [7]. Xu et al. found that *Flos populi* aqueous extract (FPAE) possesses significant anti-diarrheal activity and displays anti-microbial activity against *Salmonella* Typhi, *Escherichia coli*, and *Shigella flexneri* [6]. Ni et al. found that *Flos populi* extract has antioxidant activity and can be utilized as a potential natural antioxidant [8]. Hou et al. found three compounds in n-butanol extract of *Flos populi* that exhibited potent antioxidant capacities in vitro, and one compound displayed an exceptional inhibitory effect on NO, TNF-α, IL-6, and IL-1β production in LPS-stimulated RAW 264.7 cells [9]. However, the effect and mechanism by which FPAE prevents *S.* Pullorum infection remains unclear.

*Salmonella* virulence depends on the delivery of many effector proteins via the type III secretion system encoded by *Salmonella* pathogenicity island 1 (T3SS-1) [10]. T3SS-1 is crucial for the adhesion and invasion of non-phagocytic cells. These effector proteins have multiple activities within host cells, including fluid secretion and *Salmonella*-containing vacuole (SCV) formation [11]. After uptake by host cells, T3SS-2 is activated and furthers the infection process through its effectors [12]. In the second stage of infiltration and intracellular persistence in macrophages, T3SS-2 is an essential component.

In this study, we investigated the impact of FPAE on *S.* Pullorum growth and determined its role in the adhesion, invasion, intracellular survival, and replication of *S.* Pullorum, as well as the related mechanism. Moreover, we assessed the effect of FPAE against *S.* Pullorum in vivo and showed that it could protect chickens from infection.

## 2. Materials and Methods

### 2.1. Reagents

FPAE was purchased from Xin Cheng Da Tang Co., LTD (Xi’an, China). The concentration was 1000 mg/mL (each 1 mL was equivalent to 1 g of the original drug). Dulbecco′s modified Eagle medium (DMEM), RPMI 1640, Triton X-100, and fetal bovine serum (FBS) were purchased from Sigma-Aldrich (St. Louis, MO, USA). 

### 2.2. Bacterial Strains and Cell Culture

A virulent strain of *S.* Pullorum CVCC 519, isolated from diseased chickens in China, was purchased from the China Institute of Veterinary Drug Control (Beijing, China). The bacterium *S.* Pullorum CVCC 519 was cultured in Luria-Bertani (LB) broth or agar (Oxoid, Basingstoke, UK) at 37 °C for various time periods according to the purpose of the experiment. DF-1, a chicken embryo fibroblast cell line, was used for bacterial adhesion and invasion assays. HD11, a macrophage-like chicken cell line, was used for bacterial survival and intracellular replication assays. DF-1 cells were grown in DMEM supplemented with 10% FBS and 1% antibiotics (penicillin 100 U/mL, streptomycin 100 µg/mL), and HD11 cells were grown in RPMI 1640 medium with 20% FBS and 1% antibiotics. Both cell lines were incubated at 37 °C in a humidified atmosphere with 5% CO_2_.

### 2.3. Minimum Inhibitory Concentration (MIC), Minimum Bactericidal Concentration (MBC), and Growth Curve Assay

The MIC of FPAE was determined using the 2-fold broth microdilution reference method according to Clinical and Laboratory Standards Institute standards [13]. Briefly, 2-fold serial dilutions of FPAE were made in Mueller-Hinton (MH) broth (pH 7.2–7.4) at a volume of 100 μL per well in 96-well U-bottomed polystyrene microtiter plates. Each well was inoculated with 100 μL of CVCC 519 inoculum, yielding a final bacterial concentration of approximately 5 × 10^5^ CFUs/mL. The final tested concentrations of the anti-microbial agents ranged from 7.81 to 500 mg/mL for FPAE. The MIC was defined as the lowest concentration of the tested agent that completely inhibited visible growth in MH broth. After plating onto MH agar plates, MBC was defined as the lowest concentration of the studied agent that eliminated 99.9% of the test bacteria. In the MIC and MBC experiments, three replicates were done for each sample, and each experiment was repeated three times.

To construct growth curves, overnight cultures of *S.* Pullorum CVCC 519 inoculum were diluted to 5 × 10^5^ CFUs/mL with fresh LB broth. The suspension was supplemented with different concentrations of FPAE (62.5, 31.25, 15.62, 7.81, 3.905, 1.95, and 0.98 mg/mL) and incubated at 37 °C under continuous agitation (200 rpm). LB broth containing 0.1% DMSO was used as a negative control. During incubation, 100 μL aliquots of each sample were taken at 0, 1, 3, 5, 7, 9, 11, 13, 15, and 24 h of incubation. Each aliquot was serially diluted 10-fold, and then 100 μL was inoculated onto MH agar from at least two dilutions per time point. The samples were incubated overnight at 37 °C, and the colony counts were then determined. The corresponding data were recorded continuously to generate a growth curve. 

### 2.4. Adhesion and Invasion of DF-1 Cells

DF-1 cells were seeded in 24-well plates and cultured to 90% confluence (1 × 10^5^ cells/well) in DMEM with 10% FBS and without any added drugs in a 5% CO_2_ atmosphere at 37 °C for 24 h. After rinsing twice in serum-free medium, the DF-1 cells were inoculated with CVCC 519 (MOI = 100) and incubated at 37 °C for 2 h. Before infecting the cells, CVCC 519 (OD_600 nm_ = 0.6) was pretreated with different concentrations of FPAE (7.81, 3.905, and 0 mg/mL) for 4 h. For the adherence assay, infected cells were washed twice with PBS and lysed with sterile distilled water on ice. Bacteria in the water were plated at various dilutions to quantify the number of adherent bacteria. For the invasion assay, the infected cells were washed once with PBS and incubated with medium containing 100 μg/mL gentamycin for a further 2 h to kill the extracellular bacteria. The cells were then lysed with sterile distilled water to count viable bacteria.

### 2.5. Survival and Intracellular Replication in Macrophages

HD11 cells, a macrophage-like chicken cell line, were cultured in 24-well plates (1 × 10^5^ cells/well). The cells were infected with CVCC519 (MOI = 100) at 37 °C for 2 h after pretreatment with different concentrations of FPAE (7.81, 3.905, and 0 mg/mL) for 4 h. Finally, the cells were treated with gentamicin (100 μg/mL) for 1 h to kill extracellular bacteria.

For the survival assay, HD11 cells were lysed in sterile distilled water before serial dilution and incubation on an LB agar plate overnight at 37 °C for a bacterial count. To estimate intracellular replication, 1 mL DMEM was added to each well. After incubation for 24 or 48 h, the cells were lysed, and the appropriate dilution was plated on LB agar plates overnight at 37 °C for a bacterial count, as described in the survival assay.

### 2.6. Total RNA Extraction and Gene Expression

To assess the virulence gene expression, *S.* Pullorum CVCC 519 (10^7^ CFU/mL) was incubated in LB broth containing FPAE at different concentrations at 37 °C for 4 h. Total RNA of *S.* Pullorum CVCC 519 was extracted using an RNeasy Mini Kit (QIAGEN, Shang Hai, China). For each qRT-PCR reaction, 2 μg of total RNA was used to synthesize cDNA using HiScript II Q RT SuperMix with gDNA wiper (Vazyme, Nanjing, China). The relative gene expression was quantified via qRT-PCR using an ABI QuantStudio™ 6 Flex Real-Time PCR System (Applied Biosystems, Inc. Foster City, USA) with a SYBR Green Plus Reagent Kit (Roche, Basel, Switzerland). The primers for T3SS-1 effector (*sipA/B/C*, *sopB/E*, *invA/H*), structural (*prgH/I/J/K* and *invG*) and regulatory (*hilA*, *hilD*, and *invF*) genes are listed in Table 1. Target gene expression levels were normalized to that of *GAPDH* (reference gene; internal control) using the 2^−ΔΔCT^ method. In this experiment, three replicates were done for each sample, and the experiment was repeated three times.

### 2.7. Chicken Infection

The anti-*S.* Pullorum effect of FPAE was further determined using an infection model of Jianghan domestic chickens. All experimental protocols were approved by the Animal Care and Use Committee of the Institute of Animal Husbandry and Veterinary Medicine, Hubei Academy of Agricultural Sciences. A total of 120 one-day-old Jianghan domestic chickens tested to be free of *S*. Pullorum were randomly divided into three groups: control, *S*. Pullorum, and FPAE-treated groups. Each group had 40 chickens. The chickens were orally infected with 0.2 mL of PBS with or without 4.7 × 10^8^ CFU CVCC 519. The FPAE-treated group received intragastric administration of FPAE (4 mg/kg) once a day for 15 consecutive days starting 6 h after infection, and the control and *S*. Pullorum groups received equal volumes of PBS. The chickens were monitored twice daily for 15 days, and their body weights were recorded. Every other day, three chickens of each group were executed to monitor the colonization of the liver and spleen. The tissues and organs of the hens in each group were separated, extracted using a sterilizing tool, and weighed before homogenization at predetermined time intervals. To encourage the full discharge of intracellular microbes, the livers and spleens of each cohort were recovered in 1 mL of PBS containing 0.2% Triton X-100. To count the number of *S.* Pullorum infections in these tissues, the homogenates were serially reduced and inoculated onto LB agar containing streptomycin.

### 2.8. Statistical Analysis

All experimental data are presented as means ± standard deviation and were analyzed using GraphPad INSTAT (GraphPad Software, Inc., San Diego, CA, USA). An independent Student′s *t*-test was used to determine significant differences, and the *p* values are indicated as follows: * *p* < 0.05, ** *p* < 0.01, and *** *p* < 0.001.

## 3. Results

### 3.1. MIC, MBC, and Sub-MIC of FPAE against S. Pullorum

The MIC and MBC of FPAE against *S*. Pullorum CVCC519 were both 125 mg/mL. The growth curve showed that FPAE concentrations ranging from 15.62 to 62.5 mg/mL reduced *S*. Pullorum growth. FPAE concentrations of 7.81 mg/mL or lower showed no significant inhibitory effect on *S.* Pullorum growth (Figure 1). Therefore, concentrations of 7.81 mg/mL (1/16 MIC) and 3.905 mg/mL (1/32 MIC) were selected as sub-MICs to study the effects of FPAE on *S*. Pullorum virulence.

### 3.2. Adhesion and Invasion of S. Pullorum on DF-1 Cells

The inhibition of bacterial adhesion and invasion was assessed using a gentamicin protection assay. FPAE inhibited the adhesion and invasion of *S*. Pullorum in DF-1 cells. The FPAE inhibition efficiency against adhesion was 37.78% at 7.81 mg/mL (*p* < 0.01) and only 5.49% at 3.905 mg/mL (*p* > 0.05) (Figure 2A). The FPAE inhibition efficiency against invasion was 66.24% and 53.29% (*p* < 0.001) at concentrations of 7.81 and 3.905 mg/mL, respectively, compared to untreated *S.* Pullorum (Figure 2B).

### 3.3. Survival and Intracellular Replication of S. Pullorum in Macrophages

The inhibition of bacterial survival and replication in HD11 cells was assessed using a gentamicin protection assay. FPAE did not decrease the intracellular survival of *S.* Pullorum in HD11 cells. *S*. Pullorum replication in HD11 cells did not decrease under FPAE treatment at the concentrations shown within the 48 h observation period (Figure 3).

### 3.4. RT-qPCR Analysis of T3SS-1-Related Genes

FPAE concentrations of 7.81 and 3.905 mg/mL downregulated the transcription of these effector (*sipA/B/C*, *sopB/E*, *invA/H*) and structural (*prgH/I/J/K* and *invG*) genes compared with that of the respective control value (Figure 4A,B). We then measured some T3SS-1 regulatory genes and found that FPAE treatment also significantly reduced the expression levels of *hilA*, *hilD*, and *invF* relative to their respective control levels (Figure 4C).

### 3.5. FPAE Protects S. Pullorum-Infected Chickens in Vivo

As shown in Figure 5A, in the *S.* Pullorum group chickens began to die 1 day post infection, and the survival rate was 80% during the observation period. Among the eight deceased chickens, five in total have white fecal material pasted to the vent area. Additionally, all the infected chickens huddled near the heat source and were weak, anorectic and depressed. In contrast, chickens in the FPAE-treated group (*S.* Pullorum + FPAE 4 mL/kg) began to die 5 days after infection, and 95% of the chicks survived. The surviving chickens gradually resumed feeding after 3 days. No death occurred in the control group.

The average weight loss of chickens in the FPAE-treated group at 3 and 6 days post infection was significantly lower than that in the *S.* Pullorum group (*p* < 0.05). On day 9 after infection, the average body weight of chickens in the FPAE-treated group was equal to that in the control group (Figure 5B). Measurements in the target organs (liver and spleen) revealed a significantly higher bacterial load in the *S.* Pullorum group than in the FPAE treatment group, indicating that FPAE treatment reduced bacterial growth (Figure 5C,D). In summary, FPAE comprehensively and effectively protected chicks from *S.* Pullorum infection.

## 4. Discussion

Pullorum disease, which is caused by *S.* Pullorum, is one of the most significant infectious diseases affecting the poultry industry and is responsible for substantial global economic losses [14]. Antibiotics have been used to control pullorum disease, but their prolonged use has resulted in the rise of multidrug-resistant bacterial strains. Therefore, it is essential to identify safer and more effective means of control [15]. Although many herbs, such as *Flos populi*, have been used to treat diarrhea caused by *S.* Pullorum infection in chickens, their underlying mechanisms remain elusive.

A previous study revealed the anti-diarrheal properties of FPAE in rats and mice, as well as its anti-microbial effects against *Salmonella* Typhi, *Escherichia coli*, and *Shigella flexneri* [6]. Other studies have shown that *Flos populi* extract has antioxidant potential in vitro and in mice, as well as anti-inflammatory and analgesic activities [8,16]. In this study, we evaluated the protective effects of sub-MICs of FPAE on *S.* pullorum-infected chickens. FPAE reduced mortality, weight loss, and bacterial replication in the organs of chickens infected with *S.* Pullorum.

We then investigated the mechanism underlying FPAE inhibition of *S.* Pullorum infection, which is not fully understood. The development of infection is dependent on T3SS-1, as well as adhesin-mediated adhesion and penetration of the epithelium, which defines the capacity of *Salmonella* to penetrate host cells [17]. Once the epithelial barrier has been broken, *Salmonella* can infiltrate gut macrophages and create SCVs. The expression of T3SS-2 genes is critical for replication within macrophages. *Salmonella* can spread throughout the body, resulting in systemic infections. Here, we investigated the effect of FPAE on the adhesion and invasion abilities of *S.* Pullorum on DF-1 cells and its intracellular proliferation in macrophages. FPAE inhibited the adhesion and invasion of *S.* Pullorum in the cells but had no significant effect on intracellular proliferation. That was maybe the reason why FPAE reduced the bacterial loads in organs and decreased the mortality and weight loss of infected chickens. Therefore, we focused on T3SS-1, which determines bacterial adhesion and invasion.

T3SS-1 is a bacterial injection (needle) complex used to deliver effectors into host cells [18]. Effectors, such as SipA, SipC, SopE, SopE2, and SopB, can directly or indirectly modulate host actin dynamics to facilitate bacterial uptake [19]. The effector proteins of T3SS-1 also induce fluid secretion, cell metabolism, immune cell recruitment, and the regulation of host inflammatory responses [20]. In our study, FPAE downregulated the transcription levels of several effector genes (*sipA/B/C, sopB/E,* and *invA/H*) at a concentration that did not influence *S.* Pullorum growth. The transcription of some structural genes (*prgH/I/J/K* and *invG*) that encoded the structural proteins of the injection complex was also inhibited by FPAE. The induction of genes regulated by T3SS-1 requires the expression of its transcriptional activators, *invF* and *hilA* [21]. Furthermore, because HilA is the primary regulator of T3SS-1, the HilC-RtsA-HilD feedback loop is the most crucial core component of the regulatory networks that govern *hilA* transcription [22]. We found that the mRNA transcript levels of some regulatory genes of T3SS-1, namely *hilA*, *hilD*, and *invF*, were significantly reduced following FPAE treatment. T3SS-1 inhibitors were initially investigated in 2002 [23]. Inhibitors, including sanguinarine chloride and Csn-B, affect the secretion of T3SS-1 effectors [24,25]. Other inhibitors, including cinnamaldehyde and phenol, affect the transcription of T3SS-1 regulatory proteins [26,27]. The transcription level of T3SS-1 genes regulated by the *hilD*-*hilC*-*rstA*-*hilA* regulatory pathway is affected by the use of myricetin in *S*. Typhimurium [28]. According to the results of our study, FPAE can inhibit *Salmonella* virulence and affect the transcriptional levels of T3SS-1 genes by affecting the *hilA*-related regulatory pathway.

Our study is crucial for understanding the molecular mechanisms by which FPAE inhibits *S.* Pullorum infection. In summary, FPAE effectively inhibited the transcription of several T3SS-1-related genes, thereby impairing its ability to adhere to and invade cells. Further, FPAE reduced the bacterial loads in organs and decreased the mortality and weight loss of infected chickens. However, the key components of FPAE involved in this process and the question of how they act on T3SS-1 require further investigation.

## Figures and Tables

**Figure 1 pathogens-12-00790-f001:**
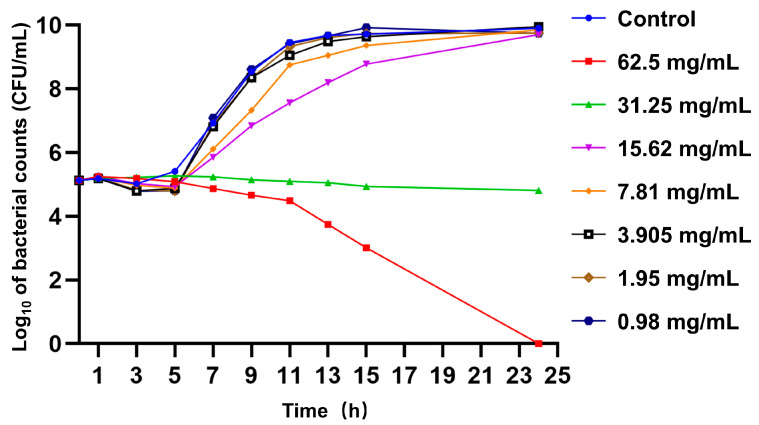
Effect of *Flos populi* aqueous extract (FPAE) on *Salmonella* Pullorum growth curve. Control: untreated *S*. Pullorum control; 62.5 mg/mL: *S*. Pullorum treated with 62.5 mg/mL of FPAE. Concentrations used were established from minimum inhibitory concentration (MIC) evaluation and represent the ½MIC of FPAE.

**Figure 2 pathogens-12-00790-f002:**
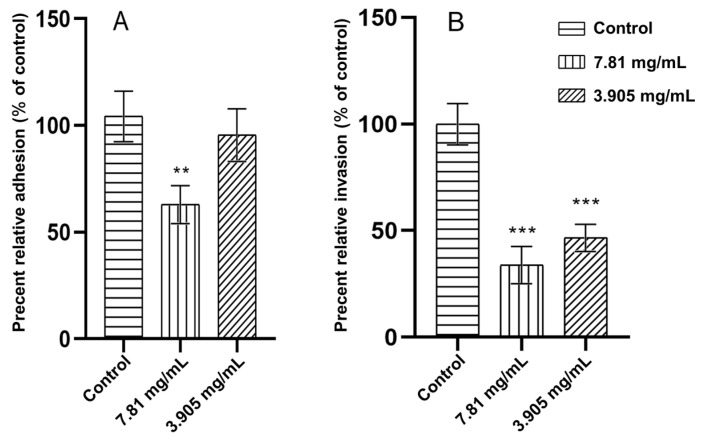
Effects of FPAE on *S*. Pullorum (**A**) adhesion to and (**B**) invasion of DF1 cells. Bars represent the means ± standard deviations (n = 3). ** *p* < 0.01 and *** *p* < 0.001.

**Figure 3 pathogens-12-00790-f003:**
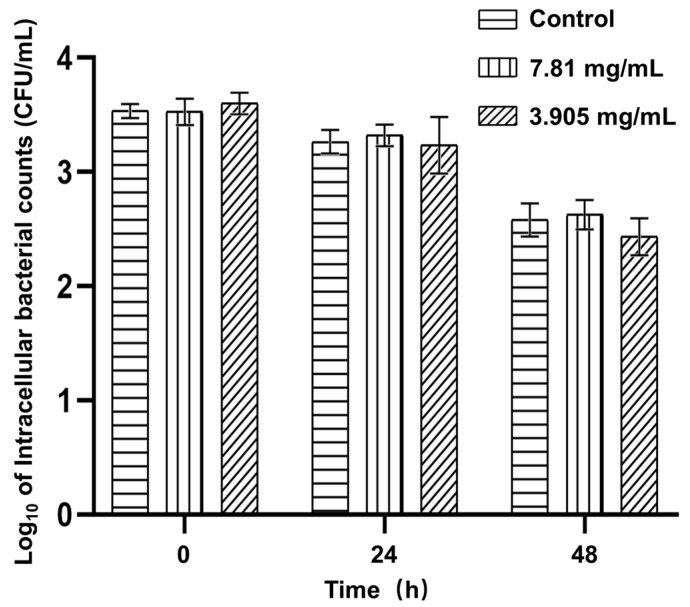
Effects of FPAE on *S*. Pullorum intracellular survival and replication within HD11 cells. Bars represent the means ± standard deviations (n = 3).

**Figure 4 pathogens-12-00790-f004:**
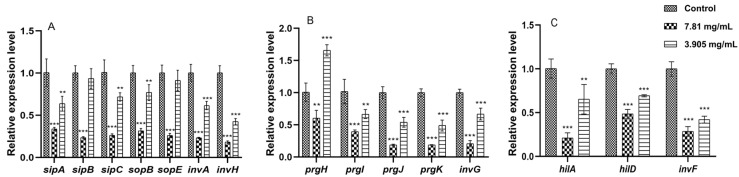
Effects of FPAE on T3SS-1-related genes of *S*. Pullorum. qRT-PCR was performed to evaluate the gene expression of (**A**) effector, (**B**) structural, and (**C**) regulatory genes. Bars represent the means ± standard deviations (n = 3). ** *p* < 0.01; *** *p* < 0.001, compared with control untreated *S*. Pullorum.

**Figure 5 pathogens-12-00790-f005:**
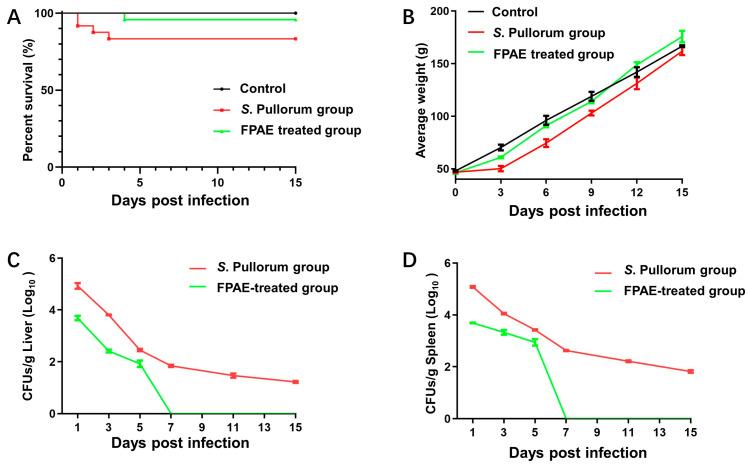
Protective effect of FPAE on *S*. Pullorum-infected chicks. (**A**) Survival curve of experimental chicks. (**B**) Effect of FPAE on the body weight of infected chicks. (**C,D**) Bacterial loading of infected chick liver and spleen. Control group: chicks infected with 0.2 mL of PBS; *S.* Pullorum group: chicks infected with 4.7 × 10^8^ CFU *S.* Pullorum CVCC 519; FPAE-treated group: chicks receiving an intragastric administration of FPAE (4 mg/kg) once daily for 15 consecutive days after infection with 4.7 × 10^8^ CFU CVCC 519. Bars represent the means ± standard deviations (n = 3).

**Table 1 pathogens-12-00790-t001:** Primer pairs used in this research.

Target Gene	Primers	Target Gene	Primers
*sipA*	5′-GCAGAGGCTGTGACCAA-3′	*prgI*	5′-TGGCGGCGTATCAGAGT -3′
	5′-GGTTACGCCATCGACTT-3′		5′-TGGCAGCATCAATATCCTTAA-3′
*sipB*	5′-CTTAAAGCCGGAACAAAG-3′	*prgJ*	5′-GGACCCTAATCTGGTGA-3′
	5′-GCGAAACATCACCCAGTAG-3′		5′-GAGCGTAATAGCGTTTC-3′
*sipC*	5′-CGACTAAAGCGAATGAGG-3	*prgK*	5′-AAAGGACTGGACCAGGAA-3′
	5′-GCAACGGCACTGGAAGA-3′		5′-AATCAGGCTCAGCAACG-3′
*sopB*	5′-GCCCTGAAGCCAGACCA-3′	*invG*	5′-CGGCTGAGCAGGTGAAT-3′
	5′-AGGCGTTGTGCGAGTTT-3′		5′-TCCATGAAGTGCCCAAA-3′
*sopE*	5′-GACATACCGACTACCCA-3′	*hilA*	5′-TCGGAAGATAAAGAGCA-3′
	5′-CTCGCATTACCTTTGAT-3		5′-TATCGCCAATGTATGAG-3′
*invA*	5′-CTTGTAGAGCATATTCGTGGAG-3′	*hilD*	5′-TTCGGAGCGGTAAACTGAC-3′
	5′-AGGAAGGTACTGCCAGAGGT-3′		5′-TTGCTGCTCGTTTGGGATA-3′
*invH*	5′-TTTACTGATCGGCTGTG-3′	*invF*	5′-GAGAATGCTGGGAGAAGAC-3′
	5′-CAATACTATTTGCGTTGG-3′		5′-AAATGTGAAGGCGATGAGT-3′
*prgH*	5′-AGCAGCCTGAGAAGTTAGA-3′	*GAPDH*	5′-CGCATCTCAGAACATCATCC-3′
	5′-TGTCCCAATTCCCAATAT-3′		5′-ACGAACGGTCAGGTCAACAA-3′

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
