# Peer review of "Flos populi (Male Inflorescence of Populus tomentosa Carrière) Aqueous Extract Suppresses Salmonella Pullorum Infection by Affecting T3SS-1"

_pathogens, 2023, doi:10.3390/pathogens12060790_

Round 1
Reviewer 1 Report
Key information on in vivo trial and replicates are missing in different assays of this study. The manuscript do not comply the requierements to assess the soundness and the reproducibility of the study.
In addition, few points in the introduction section are false and citation are inappropriated.

Author Response
Point-by-point responses to the comments of the reviewer 1
General comment:
- The general structure of the manuscript is clear and the paper is easy-to-read.
Response: Thank you for your comments.
- Several points in the introduction section have to be clarified or corrected since their meaning are unclear or inexact (see specific comments).
Response: We have corrected them in the revised manuscript (see specific comments).
- The material and methods used is globally well described except for the in vivo trials where improvements have to be performed. In fact, several essential information are lacking on the study design. For example, the number of chicken included in each experimental group is missing and the chicken breed is not mentioned. Therefore it is not possible to clearly evaluate the soundness of the study and these 2 points make the study impossible to reproduce.
Response: In this chicken infection experiment, a total of 120 one-day-old Jianghan domestic chickens tested to be free of S. Pullorum were randomly divided into three groups. Each group had 40 chickens. And we have added the information in the revised manuscript (Line 152-154).
- The absence of replicates in vitro assays (MIC/MBC and gene expression quantification) is also concerning regarding the soundness and reproducibility of the study.
Response: In the vitro assays, 3 replicates were done for each sample and each experiment was repeated 3 times. In the revised manuscript, we have included the replicate information (Line 101-102, Line 145-146).
- Knowing that, the conclusions drawn be the author, as well as the title of the manuscript, has to be nuanced.
Response: In our study, we mainly found that the inhibitory effect of FPAE on S. Pullorum infection is related to T3SS-1, so we chose this conclusion as the title. However, we described other findings with more detail in the conclusion section. (Line 288-293).
Specific comment:
- L 34-36: The number of Salmonella species is wrong. The reference cited do not refer to the white-kauffman-leMinor classification. There are 2 species of Salmonella and 6 sub-species within the species enterica.
Issenhuth-Jeanjean S, Roggentin P, Mikoleit M, Guibourdenche M, de Pinna E, Nair S, Fields PI, Weill FX. Supplement 2008-2010 (no. 48) to the White-Kauffmann-Le Minor scheme. Res Microbiol. 2014 Sep;165(7):526-30. doi: 10.1016/j.resmic.2014.07.004. Epub 2014 Jul 15. PMID: 25049166.
Response: Thank you for the recommended reference. We have corrected the number of Salmonella species and serovars in the revised manuscript (Line 34-35) and updated the reference (Line 313-314).
- L 54-55 + L 238-239: “three species of bacteria: Salmonella typhi, Escherichia coli, and Shigella flexneri” Salmonella typhi is a serovar of Salmonella not a species
Response: We have corrected them in the revised manuscript (Line 53-54, Line 247-248).
- L 59-60: in the introduction section, the author stated that the mechanism by which FPAE prevents S. Pullorum infection remains unclear. However, this effect has not been previously described (no reference cited).
Response: The effect by which FPAE prevents S. Pullorum infection, also remains unclear before we reported it. So, there is no reference cited here. We have corrected it in the revised manuscript (Line 59-60).
Reviewer 2 Report
The introduction needs to be corrected. The references mentioned are not correct. The phrases used are in the introduction of the referred works and are not results thereof. Also, lines 34-5-6 are wrong and again, the reference is not suitable there.
Line 37. SP is not strictly chicken adapted. In the past, turkeys acquired SP by incubating chicken and turkey eggs in the same hatchery. It would be important to suggest that artificial incubation favors the spread of SP. Regardless of the bird species.
The in vitro assays were done with no replicates.
The in vivo work has to be better described. The is no information about the number of birds per group. Also, was a surprise for me, challenge bird dying one day after infection. What happened with surviving birds? What was the statistical analysis method adopted?
In addition to mortality, the persistence of SP in survivors should also be evaluated, bearing in mind that this is the main mechanism for the bacteria to perpetuate itself in poultry farms. Line 194-152 – 153, should be better discussed.
Author Response
Point-by-point responses to the comments of the reviewer 2
- The introduction needs to be corrected. The references mentioned are not correct. The phrases used are in the introduction of the referred works and are not results thereof. Also, lines 34-5-6 are wrong and again, the reference is not suitable there.
Response: Thank you for the recommended reference. We have corrected the number of Salmonella species and serovars in the revised manuscript (Line 34-35) and updated the reference (Line 313-314).
- Line 37. SP is not strictly chicken adapted. In the past, turkeys acquired SP by incubating chicken and turkey eggs in the same hatchery. It would be important to suggest that artificial incubation favors the spread of SP. Regardless of the bird species.
Response: According to your suggestion, we have corrected it in the revised manuscript (Line 35-39).
- The in vitro assays were done with no replicates.
Response: In the vitro assays, 3 replicates were done for each sample and each experiment was repeated 3 times. In the revised manuscript, we have included the replicate information (Line 101-102, Line 145-146).
- The in vivo work has to be better described. The is no information about the number of birds per group. Also, was a surprise for me, challenge bird dying one day after infection. What happened with surviving birds? What was the statistical analysis method adopted?
Response: In this chicken infection experiment, a total of 120 one-day-old Jianghan domestic chickens tested to be free of S. Pullorum were randomly divided into three groups. Each group had 40 chickens. And we have added the information in the revised manuscript (Line 150-152). Four chickens in the S. Pullorum group died 1 day after infection. They huddle near the heat source, were weak before died and 1 of them have white fecal material pasted to the vent area. The chickens in the S. Pullorum group died 2 and 3 day after infection also have similar symptoms. The 5 of 8 chickens died in S. Pullorum group have white fecal material pasted to the vent area in total. The surviving chicken in FPAE-treated group gradually resume feeding after 3days (Line 219-221, 223-224). The average weight loss of chickens in the FPAE-treated group at 3 and 6 day post in-fection was significantly lower than that in the S. Pullorum group (P < 0.05) (Line 225-226). The independent Student’s t-test was used to determine significant differences.
- In addition to mortality, the persistence of SP in survivors should also be evaluated, bearing in mind that this is the main mechanism for the bacteria to perpetuate itself in poultry farms. Line 194-152 – 153, should be better discussed.
Response: In this chicken infection experiment, 3 chickens of each group were executed every other day for monitoring the colonization of the liver and spleen. And, FPAE reduced the bacterial replication in the organs. The experiments at cell level revealed that FPAE inhibited the adhesion and invasion of S. Pullorum in the cells, but had no significant effect on intracellular proliferation. That was maybe the reason. (Line 262-265).
Round 2
Reviewer 2 Report
The manuscript is fine know.
Author Response
no